# Cardiovascular Health Benefits of Exercise Training in Persons Living with Type 1 Diabetes: A Systematic Review and Meta-Analysis

**DOI:** 10.3390/jcm8020253

**Published:** 2019-02-17

**Authors:** Nana Wu, Shannon S.D. Bredin, Yanfei Guan, Kyra Dickinson, David D. Kim, Zongyu Chua, Kai Kaufman, Darren E.R. Warburton

**Affiliations:** 1Physical Activity Promotion and Chronic Disease Prevention Unit, University of British Columbia, Vancouver, BC V6T 1Z4, Canada; nana.wu@ubc.ca (N.W.); yanfei.guan@ubc.ca (Y.G.); dickinsonkyra@gmail.com (K.D.); davidd.kim@alumni.ubc.ca (D.D.K.); zongyu18@gmail.com (Z.C.); kai.kaufman@ubc.ca (K.K.); 2Laboratory for Knowledge Mobilization, University of British Columbia, Vancouver, BC V6T 1Z4, Canada; shannon.bredin@ubc.ca

**Keywords:** cardiovascular health, type 1 diabetes, exercise training

## Abstract

Exercise is advocated in the management of type 1 diabetes (T1D), but the effects of different forms of exercise training on cardiovascular risk factors in T1D still remain unclear. The aim of this study was to conduct a systematic review and meta-analysis on exercise training for cardiovascular risk factors in T1D. Six electronic databases were systematically searched for randomized or non-randomized controlled studies reporting associations between exercise training and cardiovascular risk factors in T1D. Weighted mean differences (WMD) of each cardiovascular risk factor between exercise groups and control groups were calculated using a random effects model. A total of 24 studies reported the effects of exercise training on cardiovascular risk factors. Exercise training increased maximal aerobic power (3.01 mL·kg^−1^·min^−1^, 95% confidence interval, CI, 0.94 to 6.38) and reduced glycated hemoglobin (HbA1c; −0.45%, 95% CI −0.73 to −0.17), daily insulin dosage (−0.88 U·kg^−1^, 95% CI −1.27 to −0.48), and total cholesterol (−0.38 mmol·L^−1^, 95% CI −0.71 to −0.04). Exercise training did not lead to consistent changes in body mass index (BMI), blood pressure, triglycerides, high-density lipoprotein cholesterol (HDL-C), or low-density lipoprotein cholesterol (LDL-C). In persons living with T1D, exercise training is associated with a beneficial cardiovascular profile, including improvements in lipid profile, glycemic control (decreased daily insulin dosage and HbA1c), and aerobic fitness.

## 1. Introduction

Type 1 diabetes (T1D) is an autoimmune disease characterized by insufficient production of insulin resulting from the destruction of the insulin-producing β-cells of the islets of Langerhans of the pancreas [1,2]. The prevalence of T1D continues to increase considerably. According to the latest edition of the Diabetes Atlas, more than 1.1 million children and adolescents worldwide were living with T1D in 2017, for this first time exceeding 1 million. Moreover, around 132,600 children are expected to develop T1D each year [3].

Type 1 diabetes is associated with high risks of microvascular and macrovascular complications, as well as other cardiovascular risk factors, including obesity, hypertension, hyperglycemia, dyslipidemia, insulin resistance, and physical inactivity [4,5]. Type 1 diabetes is also related to cardiovascular abnormalities (such as reduced myocardial function, increased carotid intima-media thickness, arterial stiffness, and endothelial dysfunction) that may increase the risk for the development of chronic heart failure [6,7]. Diabetic nephropathy increases the incidence of cardiovascular morbidity and mortality among individuals with diabetes. Additionally, cardiovascular disease is the most frequent cause of premature death and disability in T1D. In people between the ages of 8 and 43 with T1D, up to 5 out of 1000 people die from cardiovascular disease each year [3,8]. Accordingly, cardiovascular risk identification and prevention is essential in such a high-risk population.

Regular exercise and physical activity participation and reduced sedentary behavior are important for cardiovascular disease risk management [9,10]. Regular exercise training offers ample health benefits for persons living with T1D, resulting in improved cardiorespiratory fitness, improved vascular health, decreased insulin requirements, improved endothelial function, reduced cardiovascular disease risks, and better self-rated quality of life [10,11,12,13,14]. Exercise training has been shown to reduce the severity of cardiovascular risk factors, such as obesity and body composition, high blood pressure, lipid lipoprotein profile, and systemic inflammation [15]. Although randomized controlled trials (RCTs) examining the association between physical activity and mortality in T1D are limited, epidemiological studies suggest that regular physical activity participation reduces the risk of macrovascular disease and death [16,17]. Moreover, a systematic review and meta-analysis has shown that exercise training has an overall beneficial lowering effect on cardiovascular risk factors in type 2 diabetes mellitus [18]. However, evidence for the effects of exercise training on cardiovascular risk factors in T1D is lacking. Consequently, the aim of this study is to conduct a systematic review and meta-analysis of published randomized or non-randomized controlled studies on exercise training for cardiovascular disease risk factors in persons living with T1D. We hypothesized that exercise training would have led to significant improvements in cardiovascular risk profile in persons living with T1D.

## 2. Materials and Methods

We adhered to the standards established by the Preferred Reporting Items for Systematic Reviews and Meta-Analyses (PRISMA) recommendations [19]. This review was registered in the PROSPERO International Prospective Register of Systematic Reviews (Identifier CRD42017060953) [20]. No study protocol was published before the initiation of the systematic review and meta-analysis.

### 2.1. Search Strategy

Preliminary searches were performed to identify any existing or ongoing reviews on this topic prior to commencing this project. The systematic review was conducted to identify relevant trials by electronic searches of MEDLINE, Embase, Cochrane Central Register of Controlled Trials (CENTRAL), Web of Science, SPORTDiscus, and Cumulative Index of Nursing and Allied Health (CINAHL) from their inceptions to December 2017 (see electronic Appendix A Methods for a complete list of search terms). Individualized search strategies were designed for each database. Databases were also searched for ongoing trials using Current Controlled Trials [21] and ClinicalTrials.gov [22]. The reference lists were manually screened for all relevant additional studies and reviews. No language restrictions were imposed.

### 2.2. Study Selection

Studies that fit the following criteria were included in this review: RCTs, quasi-experimental trials, and crossover trials examining cardiovascular risk factors before and after exercise training. Cardiovascular risk factors of interest included aerobic fitness, glycated hemoglobin (HbA1c), daily insulin dosage, blood pressure, blood lipid levels, and body mass index (BMI). Exercise training modalities of interest included aerobic, resistance, and/or combined exercise and could be supervised or unsupervised. To determine chronic adaptations, the duration of training was to be no less than four weeks. Study populations included individuals of any age or sex who had been diagnosed with T1D; the comparator was the control group. Context studies reporting at least one cardiovascular disease risk factor were also considered in this review.

We excluded case studies, reviews, and studies that included persons living with type 2 diabetes or gestational diabetes, and patients with significant diabetic complications (e.g., diabetic foot, retinopathy, severe neuropathy, uncontrolled hypertension, and diabetic keto-acidosis), cardiovascular disease, or participants on lipid-lowering therapy. Studies that failed to report cardiovascular risk factors pre- and post-exercise were excluded. Two authors (N.W. and Y.G.) independently scanned titles and abstracts, and the keywords of every study identified. Both authors independently evaluated the remaining studies based on full texts, applying the eligibility criteria for included studies. Any disagreements were resolved by consensus, or by discussion with third and fourth reviewers (K.D. and D.K.). The process was overseen by a professor with expertise in systematic reviews and knowledge mobilization (S.B.).

### 2.3. Data Extraction and Quality Assessment

Two authors (N.W. and Y.G.) independently extracted data using a standardized form. If agreement was not reached regarding the extraction of the data, an additional investigator (D.W.) adjudicated the outcome. Missing data from the included studies were requested directly from the study authors. Extracted information included: authors, title of the study, year of publication, study design, study population (age, sex, diabetic duration, sample size), details of the intervention, control conditions, and outcomes.

Using the Physiotherapy Evidence Database (PEDro) scale [23], we carried out assessments of the methodological quality of each included study according to the items addressed by the tool: selection bias, performance bias, detection bias, attrition bias, and reporting bias. An independent reviewer validated the assessment process, and any discrepancies were checked by another reviewer.

### 2.4. Data Synthesis and Analysis

Review Manager software (RevMan version 5.1, Cochrane Collaboration, Oxford, UK) was used to extract data from the included studies, the primary and secondary outcome data were reported as mean ± standard deviation (SD), median (range) or weighted mean and 95% confidence intervals (CIs). We quantified and explored the statistical heterogeneity between studies using the I-squared test and chi-squared test, with 95% uncertainty intervals. Publication bias was assessed by viewing the overlap of the study CI, using funnel plot techniques. Random effects models were chosen to conduct the meta-analyses when significant heterogeneity was present. Subgroup analyses for the participants’ age, exercise frequency, type of exercise, and program duration were used to explore the sources of heterogeneity. Sensitivity analyses were conducted by excluding one study at a time to examine if the results were driven by any one study. Standard error of the mean (SEM) values were converted to SD values.

## 3. Results

### 3.1. Study Characteristics

After the removal of duplicates, 2446 articles were identified in the initial electronic search. Following screening of the titles and abstracts, 80 full articles met the eligibility criteria for further examination (Figure 1). A total of 56 articles were then excluded for the following reasons: inappropriate study design (*n* = 39); insufficient data for meta-analysis (*n* = 11); not relevant outcomes (*n* = 5); and participants with autonomic neuropathy (*n* = 1). In total, 24 controlled studies published between 1984 [24] and 2017 [25] met our inclusion criteria and examined the cardiovascular risk factors in T1D. 

Origin and setting of the included studies, characteristics of the sample, interventions, and primary outcomes assessment are summarized in Table 1. Multiple studies utilized different program durations and/or different frequencies of exercise training intervention—these studies were reported as two related trials. When accounting for differences, the total number of trials increased from 24 studies to 28 trials, each with an exercise condition and a control condition. In summary, five studies (six trials) reported diastolic and systolic blood pressure data, ten studies (ten trials) reported BMI data, five studies (six trials) reported daily insulin dosage data, 21 studies (24 trials) reported glycated hemoglobin (HbA1c) data, 11 studies (11 trials) reported peak/maximal aerobic power (VO_2peak_/VO_2max_) data, 12 studies (15 trials) reported total cholesterol and triglyceride data, 11 studies (14 trials) reported high-density lipoprotein cholesterol (HDL-C), and 9 studies (12 trials) reported low-density lipoprotein cholesterol (LDL-C).

The mean PEDro score for the 24 studies was 4.96 ± 1.71. All studies were generally of moderate quality (Table 1). However, due to the inherent problem of blinding which accounts for three of the 10 items on the PEDro checklist (eligibility criteria item does not contribute to total score), the PEDro scores often will be lower in exercise interventions of this nature (where it is not possible to blind participants to the treatment condition).

### 3.2. Participants and Exercise Intervention

Sample sizes ranged from 13 to 196, with a total of 998 participants who had been diagnosed with T1D. The duration of T1D was reported in all studies, ranging from 3.1 ± 1.7 to 24.4 ± 3.6 years (Table 1); 546 from exercise groups and 452 from control groups. Seven of the studies included adults, whereas 17 of the studies included children and adolescents.

The frequency of the exercise interventions varied between 1 and 7 times per week, with 16 of 24 studies prescribing exercise more than three times per week. Exercise intensity was reported in terms of percentage of VO_2max_ or VO_2peak_, maximum heart rate (HR_max_), or heart rate reserve (HRR). The intensity of aerobic exercise ranged between 50% and 90% VO_2max_ or VO_2peak_, 50% and 95% HR_max_, and 40% and 60% HRR. Resistance training was generally performed based on one repetition maximum value, 10 repetition maximum values, and/or as a percentage of maximal heart rate maximum (e.g., 85–95% HR_max_). Length of exercise sessions ranged between 20 and 120 min, and duration of exercise intervention ranged between six and 36 weeks. Resistance training included weight-bearing, weight training, jumping, and sprinting.

### 3.3. Aerobic Fitness

Aerobic fitness was measured in relative (to body mass) and absolute terms. Eleven studies were pooled in a meta-analysis yielding a significant effect of exercise training on relative VO_2max_ (Effect Size, ES 3.01, 95% CI 0.94 to 5.07; *p* = 0.004) [24,25,27,28,32,34,35,36,41,44,46]. However, significant heterogeneity among these studies was detected (*I*^2^ = 80%, *Q* = 50.68, *T*^2^ = 7.76, *df* = 10, *p* < 0.01).

Subgroup analysis of moderator variables (age, frequency, type of exercise, and program duration) were used to explore the sources of heterogeneity (Table 2). It was found that studies which focused on intervention with a higher frequency (≥3 times/week) had an overall treatment effect (ES 4.45, 95% CI 3.37 to 5.52; *p* < 0.001) on VO_2max_, while intervention with a lower frequency (<3 times/week), which included two studies, had limited effect (Table 2); interventions with longer duration (>12 weeks) had an overall treatment effect (ES 5.05, 95% CI 3.81 to 6.29; *p* < 0.001, Table 2) while interventions with a shorter duration (<12 weeks) had limited effect (Table 2); interventions involving aerobic exercise only had an overall treatment effect (ES 4.25, 95% CI 3.16 to 5.34; *p* < 0.001, Table 2) while interventions involving the combination of aerobic exercise and resistance training, which included two studies, had limited effect (Table 2). Sensitivity analysis showed minor shifts only, and these shifts did not impact the overall significance of the mean effect.

### 3.4. Glycemic Control

The HbA1c was measured in all included studies [2,24,25,26,27,28,29,30,31,32,33,34,35,36,37,38,39,40,41,42,43,44,45,46]. Twenty-one studies (24 trials) were appropriate for meta-analysis [2,24,25,26,27,28,29,30,31,32,33,34,35,36,37,39,41,42,43,45,46]. Statistically significant differences in the reduction of this parameter were found in five studies favoring the exercise intervention [2,24,26,30,31]. In T1D individuals, there was a statistically significant decrease in mean HbA1c in exercise trials compared to control trials (ES −0.45%, 95% CI −0.73% to −0.17%; *p* = 0.001) (Table 2). Heterogeneity was found to be high between these studies (*I*^2^ = 76%; *Q* = 97.69, *T*^2^ = 0.24, and *df* = 23).

Subgroup analyses of moderator variables (age, frequency, type of exercise, and program duration) were used to explore the sources of heterogeneity (Table 2). There was a statistically significant reduction in HbA1c of 0.60% (95% CI −1.07% to −0.14%) in children and adolescents. Despite the fact that heterogeneity remained significant (*I*^2^ = 74%; *Q* = 61.03, *T*^2^ = 0.58, and *df* = 16), no effect was seen in the adult studies. Studies focused on interventions with a higher frequency (≥3 times/week) had an overall treatment effect (ES −0.53%, 95% CI −0.88% to −0.17%; *p* = 0.004) on HbA1c while interventions with a lower frequency (<3 times/week) had a lesser effect (Table 2); interventions with longer duration (>12 weeks) had an overall treatment effect (ES −0.56%, 95% CI −0.95% to −0.17%; *p* = 0.005, Table 2) while interventions with shorter duration (≤12 week) had no effect (*p* = 0.38, Table 2); and interventions involving aerobic exercise only had no treatment effect (*p* = 0.09, Table 2) while interventions involving the combination of aerobic exercise and resistance training had an overall treatment effect (ES −0.56%, 95% CI −1.05% to −0.08% *p* < 0.01, Table 2). Sensitivity analysis showed minimal shifts only, which did not influence the overall significance of the mean effect.

### 3.5. Daily Insulin Dosage

The weighted mean treatment effect of the six trials which measured daily insulin dosage was −0.88 U·kg^−1^·day^−1^ (95% CI −1.27 to −0.48; *p* < 0.001; Table 2) indicating a decrease in daily insulin requirements in diabetic individuals who participated in an exercise training program [2,30,32,35,43]. Heterogeneity was found to be high between these studies (*I*^2^ = 98%; *Q* = 276.52, *T*^2^ = 0.22, and *df* = 5).

There was a statistically significant reduction in daily insulin requirements of 1.69 U·kg^−1^·day^−1^ (95% CI −2.43 to −0.95; *p* < 0.001; Table 2) in children and adolescents despite the fact that heterogeneity remained significant (*I*^2^ = 99%; *Q* = 256.38, *T*^2^ = 0.52 and *df* = 3). No effect was seen in adult studies, with only two included studies. Sensitivity analysis showed minor shifts only, and these shifts did not affect the overall significance of the mean effect.

### 3.6. Lipid Profiles

Serum lipids were measured in 12 studies [2,26,28,31,32,33,35,39,42,43,44,46]. Among these studies, total cholesterol, triglycerides, HDL-C, and LDL-C levels were measured. Fifteen trials reporting total cholesterol and triglycerides, 14 trials reporting HDL-C, and 12 trials reporting LDL-C were pooled for meta-analysis. The pooled effect of the exercise training intervention was a statistically significant reduction in total cholesterol of 0.38 mmol·L^−1^ (95% CI −0.71 to −0.04; *p* = 0.03; Table 2). Heterogeneity was found to be high between these studies (*I*^2^= 89%; *Q* = 125.08, *T*^2^ = 0.33, and *df* = 14).

Subgroup analyses of moderator variables (age, frequency, type of exercise, and program duration) were used to explore the sources of heterogeneity (Table 2). A greater total cholesterol reduction was seen in the seven trials of children and young adults [2,26,28,33,43], which was considered significant (ES −0.84, 95% CI −1.22 to −0.46; *p* < 0.01). In the eight trials of adult studies, no statistically significant effect was seen (ES −0.02, 95% CI −0.25 to 0.21; *p* = 0.86) [31,32,35,39,44,47].

There were no other moderator variables found influencing the variability among studies examining total cholesterol. Sensitivity analysis revealed that a study by Salem et al. [2] in the subgroup “children and young adults” influenced the results. The removal of this study changed the “children and young adults” subgroup ES to −0.45 (95% CI −0.70 to 0.19, *p* < 0.01) and overall ES to −0.15 (95% CI −0.36 to 0.05, *p* = 0.14), removing its significance.

The meta-analysis of the random-effects model revealed a small mean effect for exercise to decrease triglycerides values (ES −0.09, 95% CI −0.19 to 0.01, *n* = 15), although this is only trended towards a significant difference (*p* = 0.08). There was high heterogeneity among these studies (*I*^2^ = 83%; *Q* = 82.66, *df* = 14, *p* < 0.01). Sensitivity analysis showed that removing a study with the largest positive ES by Campaigne et al. [28], influenced the results. The removal of this trial changed the ES to −0.11 (95% CI −0.21 to −0.01) and would also create a significant difference (*p* = 0.03).

A significant effect of exercise on HDL-C (ES −0.03, 95% CI −0.20 to 0.14; *n* = 14; *p* = 0.74) or LDL-C (ES −0.03, 95% CI −0.14 to 0.09; *n* = 12; *p* = 0.63) was not found. Sensitivity analysis showed minimal shifts only, which did not influence the overall significance of the mean effect.

### 3.7. Body Mass Index, Blood Pressure, and Quality of Life

Exercise was not found to have a significant effect on BMI (ES −1.00; CI −2.19 to 0.18; *p* = 0.10). Heterogeneity was found to be high between these studies (*I*^2^ = 88%; *Q* = 74.60, *T*^2^ = 3.03, and *df* = 9). Sensitivity analysis showed that a study by Roberts et al. [45] influenced the results. The removal of this trial changed the ES to −1.31 (CI −2.54 to −0.09) and also would create a significant difference (*p* = 0.04).

Out of five controlled intervention studies [2,25,27,38,39], three studies detected improvements with respect to systolic or diastolic blood pressure [2,25,27], while two studies did not [38,39]. No significant relationships were found between exercise and changes in systolic blood pressure (ES 6.10, 95% CI −0.58 to 12.78), or diastolic blood pressure (ES 0.54, 95% CI −1.29 to 2.36; *p* = 0.57). No significant heterogeneity among diastolic blood pressure studies was detected (*I*^2^ = 43%; *Q* = 8.71, *df* = 5, *p* = 0.12). Sensitivity analysis showed minimal shifts, which did not influence the overall significance of the mean effect.

Quality of life was measured in three studies using three different survey measures (i.e.; the Short Form (36) Health Survey (SF-36), Diabetes Quality-of-life (DQOL), and Subjective Quality of Life (SQOL)) [29,33,38]. One study reported a positive effect on quality of life in the exercise group [33]. There was insufficient data for pooling meta-analysis.

### 3.8. Adverse Events

The frequency of hypoglycemia was reported in seven studies [2,27,29,36,41,44,46]. One study reported frequent hypoglycemia episodes during and after exercise in the exercise group [29]. One study reported an increase in hypoglycemic symptoms during the first two weeks of the exercise program, but the frequency of hypoglycemic attacks declined thereafter [46]. Collectively, the incidence of adverse exercise-related events was low.

## 4. Discussion

This systematic review and meta-analysis included 24 studies examining the effects of exercise training on cardiovascular disease risk factors in persons living with T1D, indicating evidence for clinically important health benefits of exercise training compared to no exercise intervention on various cardiovascular risk factors. More specifically, the results of our meta-analysis indicate significant effects of exercise training on enhancing aerobic fitness (VO_2max_) while decreasing HbA1c, daily insulin dosage, and total cholesterol. However, no significant difference was found with respect to BMI, blood pressure, triglycerides, HDL-C, or LDL-C. Collectively, these findings reinforce the importance of routine exercise participation in secondary T1D management to delay and/or reduce the risk of cardiovascular disease.

### 4.1. Aerobic Fitness

Aerobic fitness is related inversely to cardiovascular disease risk and all-cause mortality in T1D [47]. The gold standard assessment of aerobic fitness is VO_2max_. Eleven studies measured VO_2max_ or VO_2peak_, revealing a significant increase of 3.01 mL·kg^−1^·min^−1^ in patients living with T1D who followed a structured exercise training program. However, it should be noted that changes in VO_2max_ or VO_2peak_ (expressed in relative terms) may be affected by the changes of body weight. In this systematic review, exercise training improved aerobic fitness in T1D without significant changes in BMI. This finding further supports the potential for exercise training to improve cardiovascular health and profile independent of changes in body composition [10,13,14]. In clinical terms, several authors have recently demonstrated the importance of similar changes in aerobic fitness for reducing the risk for premature mortality [48,49]. For instance, Martin and colleagues revealed that each metabolic equivalent of task (MET, approximately 3.5 mL·kg^−1^·min^−1^) increase in aerobic fitness was associated with a 25% reduction in all-cause mortality [48]. Similarly, Kokkinos and colleagues revealed that there was a 12% lower risk for premature mortality for each 1-MET increase in exercise capacity [49].

Our sub-analyses showed that greater aerobic fitness improvements were generally attained from interventions that involved aerobic exercise with a higher frequency per week, and/or a longer duration. However, there was existing evidence of beneficial changes with rather small volumes of exercise [50]. This research demonstrates that a one-size fits all approach to exercise prescription is not ideal for persons living with T1D [10]. Further research is required to establish the minimal and optimal levels of exercise training for changes in aerobic fitness. Moreover, additional research is required to determine the effects of exercise training on other determinants of health-related physical fitness (such as musculoskeletal fitness).

### 4.2. Glycemic Control

Glycemic control is strongly associated with cardiovascular disease and is fundamental to diabetes management. For each percentage point increase in mean HbA1c, the risk of cardiovascular disease increases by 31% [51]. In our current systematic review, the majority of included individual studies on exercise training demonstrated no significant results on glycemic control. This is possibly due to insufficient power to detect a difference in a small sample of patients in each individual study. However, when viewed in totality, our meta-analysis of the grouped studies shows significant effects of exercise on reduction of HbA1c, reinforcing the importance of exercise in the clinical diabetic management to improve glycemic control. Improving glycemic control is important for decreasing cardiovascular disease and correlates with reduced cardiovascular disease-related mortality in an epidemiologic and longitudinal study [51].

Our sub-analysis showed that training less than three times a week or less than 12 weeks in duration may not be enough to improve HbA1c in T1D. It also showed that exercise training has greater beneficial effects when it involved a combination of aerobic exercise and resistance training. Salem et al. and Aouadi et al. found that increased frequency and a longer period of exercise training resulted in greater reductions of HbA1c [2,26]. Therefore, for optimal reductions in HbA1c it is recommended that combined aerobic and resistance training be performed more than 3 times/week for more than 12 weeks in duration. Interpretation of these finding should be considered with caution as some unclear risk of bias and significant heterogeneity were present in most of the included studies, with the additional unknown confounding effect of baseline HbA1c levels, diet, hypo/hyperglycemic episodes, and insulin dose in treatment practices.

### 4.3. Daily Insulin Dosage

Insulin resistance leads predominant in the pathophysiology of cardiovascular disease in type 2 diabetes and has recently emerged as a consistent finding among contemporary youth with T1D [52]. Insulin resistance is a strong risk factor for cardiovascular disease in individuals with T1D [52]. Insulin dose adjustment in T1D is preferentially based on blood glucose monitoring in order to avoid hypoglycemia associated with exercise [12]. The results of this meta-analysis demonstrate that a fall of up to 0.88 U·kg^−1^·day^−1^ in insulin requirements in T1D was induced by exercise. It is possible that these reductions masked the glycemic control improvement as measured by HbA1c [2]. In T1D, there is an increased risk of hypoglycemia following exercise. Reducing the insulin requirements and increasing carbohydrate intake before exercise are suitable approaches to prevent exercise-induced hyperglycemia [12].

### 4.4. Lipid Profiles

Previous study has shown that approximately 15% of children with T1D have higher levels of LDL-C and triglycerides, in particular LDL-C, which is a well-established risk factor for cardiovascular disease [52]. Results show that exercise decreases total cholesterol levels by 0.38 mmol·L^−1^ in T1D patients. Total cholesterol levels changes are associated with decreased risk of heart disease [53]. However, we did not find statistical support for the existence of a relationship between exercise and improved triglycerides, HDL-C and LDL-C, among individuals with T1D. Higher exercise frequency and longer duration of exercise engagement were found to be significantly associated with total cholesterol improvement. Our findings were consistent with previous reports by Salem et al. who found that frequent exercise was associated with a statistically significant decrease in the lipid profile levels of total cholesterol in T1D [2]. Moreover, Aouadi et al. found that increasing frequency and duration of exercise intervention were associated with lower triglycerides, with LDL-C, and HDL-C improvement in T1D [26]. Therefore, it is important for persons living with T1D to engage in regular exercise.

### 4.5. Body Mass Index, Blood Pressure, and Quality of Life

Overweight and obesity are very common in children with T1D and major influences on the development of cardiovascular disease [54]. No significant relationships were found in this meta-analysis between exercise training and changes in BMI. The majority of the participants were lean individuals (BMI < 25 kg·m^−2^), and only three of the studies evaluated overweight and obese individuals (BMI > 25 kg·m^−2^) [27,32,33]. It can be inferred that those individuals with higher BMI at study entry may experience a greater improvement in BMI with exercise training than leaner individuals. Further research in this field is warranted.

Hypertension is a well-established risk factor for cardiovascular disease and more prevalent in people with T1D than in the general population [55]. Treatment of high blood pressure is one of the most important strategies to prevent cardiovascular disease in T1D patients. The meta-analysis was unable to identify if exercise training contributes to a significant change in blood pressure. However, a meta-analysis of 54 randomized trials evaluated 2419 participants and found that aerobic exercise reduces blood pressure in both hypertensive and normotensive individuals [56]. Moreover, in a randomized controlled trial about effects of exercise intensity on blood pressure in type 2 diabetic patients, it was shown that higher intensity exercise may elicit greater reductions in blood pressure than lower intensity exercise [57]. Therefore, it is likely that exercise interventions prescribing higher levels of exercise quantity need to be carried out to positively affect blood pressure in persons living with T1D. Further research is required to fully elucidate the effects of exercise training on blood pressure in persons living with T1D.

Previous research highlights the importance of quality of life in T1D as this outcome can often be poorer than in peers without diabetes [58]. The effects of exercise training on quality of life were well documented in type 2 diabetes [59], but the body of evidence for T1D is very limited. A systematic review has shown that aerobic exercise training was a safe and effective way to improve the quality of life in persons living with type 2 diabetes, and patients can feel more content, motivated, and confident as a result of professionally supervised, group-based exercise training [59]. However, this has not been directly investigated in persons living with T1D. Previously, D’hooge et al. reported that exercise improves quality of life in persons living with T1D and this improvement is generally associated with lower quality of life at baseline [29]. Further research in this field is warranted.

### 4.6. Adverse Events

Recent work has reinforced the importance of exercise training for persons living with chronic medical conditions (including T1D) with the benefits of exercise consistently outweighing the risks [10,13,14,57]. However, in comparison to other medical conditions, leading experts have outlined the importance of careful monitoring of exercise-related risks (in particular hypoglycemia) [60]. Hypoglycemia is associated with an increased risk of cardiovascular events and all-cause mortality in insulin-treated patients with diabetes [61,62,63]. A retrospective analysis of a large cohort of individuals with T1D on continuous subcutaneous insulin infusion pointed to a higher prevalence of cardiovascular disease in those with repeated severe hypoglycemia [64]. Patients with T1D and their parents often avoid engaging in exercise training due to a fear of hypoglycemia [65]. However, it should be noted that our systematic review revealed a relatively low incidence of exercise-related adverse events. The vast majority of patients with T1D tolerated the training well, and the frequency and intensity of hypoglycemic reactions did not change during the exercise training program in most included studies [2,27,29,36,41,44,46]. D’hooge et al. reported that frequent hypoglycemia episodes during and after exercise [29]. Yki-Jarvinen et al. showed that the minimal increase in hypoglycemia could easily have been avoided through insulin adjustment [46]. Exercise-induced hypoglycemia may be a result of blunted glucagon response, reduced adrenomedullary response, and diminished clearance of injected insulin [12]. In addition, some people with T1D with poor glycemic control may have low hepatic glycogen content, which may also contribute to exercise-induced hypoglycemia [66]. Collectively, this research supported the belief that the benefits of exercise training far outweigh the risks in persons living with T1D [10,13,14,60].

### 4.7. Management Options

It is important to acknowledge the recent advancements in T1D treatment strategies that may affect the findings of this systematic review. Eleven of 24 studies were published before the year 2000, and the remaining studies were published after the year 2000. Most of the studies focused on the effects of long-term exercise intervention in T1D, offering limited information about the treatment therapy. Multiple daily injection therapy and continuous subcutaneous insulin infusion (e.g., insulin pumps) are effective management approaches that have been increasingly used by persons living with T1D to help maintain more normal glucose levels. Continuous subcutaneous insulin infusions have evolved significantly since their introduction in the 1970s and offer the capacity to modify basal infusion rate and to obtain an effect in 1–2 h [60]. Continuous subcutaneous insulin infusion has been in particular associated with improvements in glycemic control [67].

Blood glucose level fluctuations are challenging to manage before, during, and after exercise in T1D. Thus, glycemic management is based on frequent glucose monitoring, insulin dosage adjustments, and carbohydrate intake modification before, during, and after exercise [60]. From the current findings, it is not completely clear what form of glucose assessment (e.g., self-monitored vs. continuous glucose monitoring), insulin dosage modification, or carbohydrate adjustments were used. As such, the potential confounding effects of these treatment strategies should not be overlooked. Accordingly, further research is warranted to determine whether the incidence of adverse exercise-related events varies in individuals living with T1D according to the treatment strategy employed.

### 4.8. Hyperglycemia, Dyslipidemia, and Obesity

Hyperglycemia is associated with adverse cardiovascular outcomes including vascular smooth muscle dysfunction in women with T1D, increased carotid intimal medial thickness, and impaired diastolic velocities in youth with T1D [4]. The Diabetes Control and Complications Trial/Epidemiology of Diabetes Interventions and Complications study demonstrated that achieving an HbA1c of <7% reduced the incidence of microvascular complications of T1D compared intensive versus standard glycemic control during a 6.5-year period [51]. After an average follow-up of 17 years, the intensive glycemic control therapy was associated with a 57% reduction in major cardiovascular disease outcomes even with deterioration in glucose control [51]. The American Diabetes Association recommends that individuals with diabetes control HbA1c values less than 7% to lower the risk of developing diabetes-related complications [68].

Dyslipidemia is a risk factor for cardiovascular disease in people with diabetes. Lipid levels in T1D are associated with cardiac and vascular abnormalities, suggesting direct effects of lipids on cardiovascular function including abnormal plethysmography responses, endothelial dysfunction, carotid intimal medial thickness, and aortic intimal medial thickness all correlated independently with LDL-C in youth with T1D [4]. The American Diabetes Association recommends target LDL-C levels for adults with diabetes of <100 mg·dL^−1^ (2.60 mmol·L^−1^); HDL-C levels of >40 mg·dL^−1^ (1.02 mmol·L^−1^); and triglyceride levels of <150 mg·dL^−1^ (1.7 mmol·L^−1^) [69]. Regular physical activity, severe dietary fat restriction (<10% of calories), and pharmacological therapy are recommended to reduce the risk of dyslipidemia, pancreatitis and cardiovascular diseases [69].

Obesity is an important risk factor for cardiovascular diseases. One study found that individuals with T1D who received the supra-physiological insulin doses had increased weight gain and worse total cholesterol and LDL-C, central obesity, insulin resistance, blood pressure, more coronary artery calcifications, and higher carotid intima-media thickness, underlying the role of obesity in contributing to cardiovascular disease in persons with T1D [51]. However, Gruberg et al. described better outcomes in overweight and obese patients with coronary heart disease undergoing percutaneous coronary intervention compared with their normal-weight counterparts as an “obesity paradox” [70]. Recently, a large cohort of almost 300,000 white European adults by Iliodromiti and colleagues challenged the “obesity paradox”, showing that risk of cardiovascular disease (such as heart attacks, strokes, and high blood pressure) increases as BMI increases [71]. A systematic review and meta-analysis found that exercise intervention could improve cardiovascular health by reducing BMI, HbA1c, triglycerides, and total cholesterol in youth with T1D [72], which indicates that exercise can be effective in reducing the risk of cardiovascular disease in youth with T1D.

### 4.9. Comparison with Existing Literature

The results of this meta-analysis are broadly similar to those of previous reviews. In MacMillan et al.’s meta-analysis of trials of unsupervised exercise and alternative forms of exercise (i.e.; Pilates) vs sedentary behavior intervention in youth with T1D, effect sizes for HbA1c (ES −0.85%, 95% CI −1.45% to −0.25%) more strongly favored exercise [73]. Quirk et al.’s [72] meta-analysis, which included both controlled and uncontrolled (pre and post) trials of physical activity interventions in children and young people with T1D until 2014, reported a small effect size of exercise training on HbA1c (ES −0.52%, 95% CI −0.97% to −0.07%), BMI (ES −0.41, 95% CI −0.70 to −0.12), triglycerides (ES −0.70, 95% CI −1.25 to −0.14), and total cholesterol (ES −0.91, 95% CI −1.66 to −0.17). Our meta-analysis analyzed patients of all ages as well as four additional outcome variables not included in previous meta-analyses: systolic blood pressure, diastolic blood pressure, HDL-C, and LDL-C, as well as identifying trials with control groups. Tonoli et al.’s meta-analysis reported a significant but small HbA1c-lowering effect of exercise in T1D (ES −0.27%, 95% CI −0.06 to −0.47), and subgroup analyses found that trials restricted to T1D with poorly controlled HbA1c youth had greater overall treatment effect (ES −0.66%, 95% CI −0.99 to 0.34) [74]. These results suggest that exercise training could decease the HbA1c level, and participants with poor glycemic control before intervention may experience a greater reduction in HbA1c with exercise training.

However, Kennedy et al.’s meta-analysis with a focus on glycemic control included randomized and non-randomized controlled trials, finding no changes with exercise training [75]. Ostman et al.’s meta-analysis found exercise training improves body mass, BMI, VO_2max_, and LDL in adults and insulin dose, waist circumference, LDL and triglycerides in children with T1D [76]. However, this meta-analysis used different inclusion criteria, including only RCT studies. Our meta-analysis included studies of quasi-experimental trials and crossover trials that were excluded in Ostman et al.’s meta-analysis [25,26,28,31,38,41,42,45,46]. We believe these differences accounts for the partially inconsistent conclusion. Ostman et al. [76] did however agree that studies of exercise training showed improvement in cardiovascular risk factors in T1D.

### 4.10. Strengths and Limitations

It was a goal to minimize selection bias through a comprehensive literature search and perform a comprehensive systematic literature review. A large number of RCTs relevant to our meta-analysis were included. Subgroup analyses were conducted to assess the effects among these participants of different age groups, exercise interventions of various frequency, types of exercise, and program durations. The applicability of the results was assessed.

Whilst this meta-analysis provides useful updated information for healthcare providers and policy-makers, the results should be considered with the following limitations. First, most meta-analyses had substantial to considerable heterogeneity. Our systematic subgroup analyses attempted to identify reasons for heterogeneity and reduced heterogeneity. A few subgroup analyses still showed large heterogeneity, potentially indicating the inadequate definition of subgroups, with substantial differences in intensity of interventions, gender of patients, control conditions, and/or outcome assessments. All potential explanations would reduce the confidence in the effects found in this meta-analysis. Second, several studies did not fully report outcomes of interest, and not all authors who were contacted for further information to provide missing data responded. Additionally, for many studies, the standard deviation of the post-intervention outcome was not reported, and this data was inputted using the mean correlation coefficient from the available data. Sensitivity analyses were performed using a range of correlation coefficients, which did not show any changes to the pooled effect sizes.

### 4.11. Future Research

Specificity of timing, frequency, duration, and intensity of exercise intervention is required to see beneficial results for cardiovascular risk factors and to provide evidence-based activity recommendations specifically designed for T1D. Growing evidence indicates that generic physical activity guidelines are likely not optimal for persons living with chronic medical conditions. In a recent systematic review of current systematic reviews, Warburton and Bredin revealed that marked health benefits are achieved in persons living with chronic medical conditions (including diabetes) with relatively minor volumes of physical activity [10]. This research directly challenged current threshold-based messaging related to physical activity and health. Future research should more carefully examine the minimal and optimal dosages of physical activity for health benefits in persons living with T1D. In particular, additional research is warranted that more closely examines the relationships between physical activity volumes (and/or intensities) and glycemia control, insulin dosages, aerobic fitness, and quality of life. Furthermore, more studies with larger sample sizes are required to examine the relative contributions of multicomponent interventions and differential glycemic control treatment strategies.

## 5. Conclusions

Our findings support the conclusion that exercise training plays an important role in the prevention of cardiovascular disease in T1D. The meta-analysis reveals that exercise training may result in positive changes in biological cardiovascular risk factors including aerobic fitness, HbA1c, insulin dosage, and lipids in persons living with T1D. However, these effects were not clearly distinguishable from heterogeneity and bias. The optimal and minimal dosages of exercise for beneficial changes in the cardiovascular risk profile of those living with T1D remain to be determined.

Further examination of diet and hypo/hyperglycemic episodes should be executed to fully understand the benefits of exercise on persons living with T1D. The results and recommendations of this meta-analysis are of relevance for health professionals involved in the primary prevention of cardiovascular disease, highlighting further the importance of exercise as a cornerstone in diabetes management and health promotion in T1D. Further research is required in order to demonstrate the full benefits of exercise training for patients of all ages living with T1D.

## Figures and Tables

**Figure 1 jcm-08-00253-f001:**
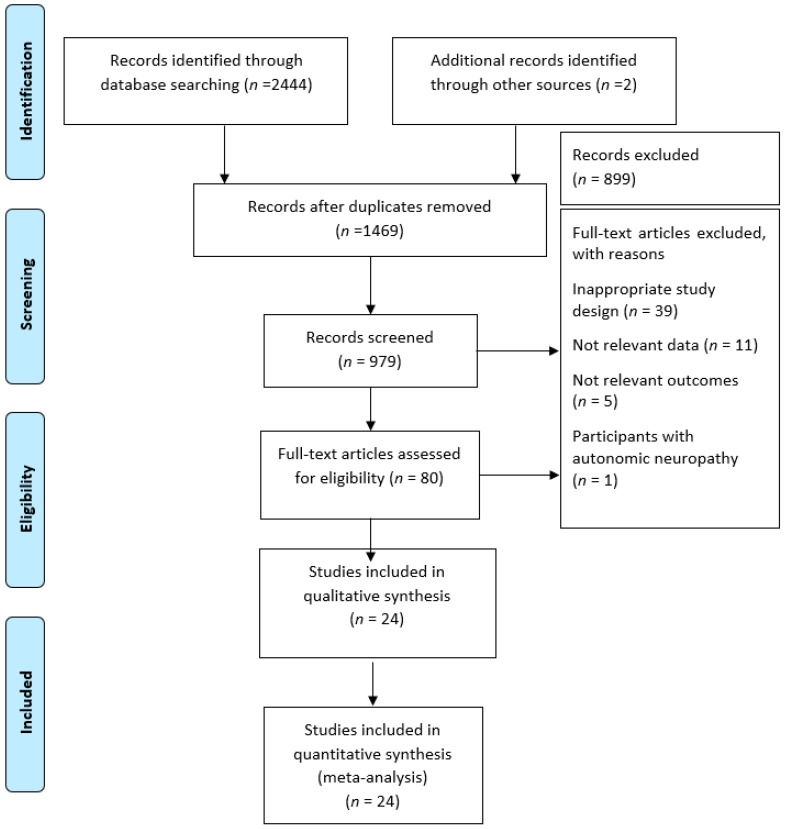
Study flow diagram in line with Preferred Reporting Items for Systematic Reviews and Meta-Analyses (PRISMA) recommendations.

**Table 1 jcm-08-00253-t001:** Characteristics of the included studies.

Study, Year (Ref.)	Participants (Age, Simple Size, Diabetes Duration)	Intervention	Outcome Measures	PEDro Score
*N*	Mean Age ± SD/Age Range (Years)	Duration of T1D (Years)	(Exercise Frequency, Intensity, Type of Exercise, Time Duration per Session)	Program Duration
Aouadi et al. 2011 ^a^ [26]	EG: 11 CG: 11	EG: 12.2 ± 1.5; CG: 12.9 ± 1.3	EG: 3.6 ± 0.8; CG: 3.2 ± 0.54	EG: 2×/week, 50–55% HR_max_ at weeks 1–2; 55–60% HR_max_ at weeks 3–4; 60–65% HR_max_ at weeks 5–24; 10–15 min WU and CD + 40–50 min aerobic exercise CG: continued with normal PA behavior	24 weeks	BMI  in both groups HDL-C ↑ significantly in EG Triglyceride ↓ significantly in EG Daily insulin dosage ↓ in EG	6
Aouadi et al. 2011 ^b^ [26]	EG: 11 CG: 11	EG: 13.5 ± 0.8; CG: 12.9 ± 1.3	EG: 4.1 ± 1.3; CG: 3.2 ± 0.54	EG: 4×/week, 50–55% HR_max_ at weeks 1–2; 55–60% HR_max_ at weeks 3–4; 60–65% HR_max_ at weeks 5–24; 10–15 min WU and CD + 40–50 min aerobic exercise CG: continued with normal PA behavior	24 weeks	BMI  in EG, CG HbA1c ↓ significantly in EG HDL-C ↑ significantly in EG Triglycerides ↓ significantly in EG LDL-C ↓ significantly in EG Daily insulin dosage ↓ in EG	6
Brazeau et al. 2014 [27]	EG: 23 (12 F; 11 M) CG: 25 (14 F; 11 M)	EG: 45.1 ± 14.5; CG: 44.2 ± 12.5	EG: 20.3 ± 12.9; CG: 24.4 ± 13.6	EG: leaflet on PA promotion with exercise 1×/week, 60 min of various activities (endurance, resistance, flexibility) + 30 min of counselling (PA and introduced glycemic management related to PA) CG: received leaflet on PA promotion and carried out normal PA	12 weeks	Weight ↓ significantly in EG BMI  in both groups HbA1c  in both groups VO_2peak_ ↑ significantly in EG SBP ↓ significantly in EG DBP ↓ significantly in EG	7
Campaigne et al. 1984 [24]	EG: 9 CG: 10	EG: 9.0 ± 0.47 (SEM); CG: 8.5 ± 0.57 (SEM)	EG: 5.1 ± 0.95 (SEM); CG: 3.89 ± 0.70 (SEM)	EG: 3×/week, HR ≥ 160 bpm, 30 min of vigorous exercise (running, movement to music), supervised aerobic exercise CG: continued with normal PA behavior	12 weeks	Weight ↑ significantly in EG Weight ↑ in CG FBG ↓ significantly in EG, significantly lower than CG HbA1c ↓ significantly in EG, significantly lower than CG Daily insulin dosage  in both groups VO_2peak_ ↑ significantly in EG	5
Campaigne et al. 1985 [28]	EG: 9 (6 F; 3 M) CG: 5 (3 M; 2 F)	EG: 16.0 ± 1 (SEM); CG: 15 ± 0.4 (SEM)	EG: 6.6 ± 1.1 (SEM); CG: 6.2 ± 1.1 (SEM);	EG: 3×/week, HR > 160 bpm intensity, 10 min WU + 25 min aerobic movement to music + 10 min CD, supervised aerobic exercise CG: continued with normal PA behavior	12 weeks	Weight  in both groups Lean body mass ↑ in EG,  in CG VO_2peak_ ↑ significantly in EG,  in CG LDL-C ↓ significantly in EG,  in CG Total cholesterol, triglyceride, HDL-C  in both groups HbA1  in both groups Daily insulin dosage  in both groups	5
D’ hooge et al. 2011 [29]	EG: 8 CG: 8	EG: 14.1 (10.1–16.8); CG: 13.2 (10.1–15.3)	EG: 5.4 (3.4–7.3); CG: 5.3 (2.9–5.9)	EG: 2×/week, 60% HRR increased to 70% HRR after 6 weeks, 75% HRR after 12 weeks; 5 min WU + 30 min strength training (upper limbs, lower limbs, and abdominal muscles, 10 min each) + 30 min aerobic training (cycling, running and stepping, 10 min each) + 5 min CD. Strength training: first 12 sessions: 2 sets of 15 reps at 20 RM; next 12 sessions: 2 sets of 12 reps at 17 RM; final 8 weeks: 3 sets of 10 reps at 12 RM; 60 s rest between 2 sets; supervised aerobic exercise and strength training CG: normal activity	20 weeks	Weight  in both groups BMI  in both groups VO_2peak_  in both groups Muscle fatigue score, number of sit to stand, upper and lower limb strength, 6 min walking distance ↑ significantly in EG HbA1c  in both groups FBG  in both groups Daily insulin dosage ↓ significantly in EG,  in CG Quality of life (SF-36)  in both groups	7
Dahl-Jorgensen et al. 1980 [30]	EG: 14 CG: 8	9–15	5	EG: 2×/week, 60 min; supervised exercise; and supplemented by a weekly home exercise experience CG: did not participate in any standardized exercise regime	20 weeks	VO_2peak_  in both groups Daily insulin dosage  in both groups HbA1c ↓ significantly in EG,  in CG	5
Durak et al. 1990(crossover) [31]	EG: 8 CG: 8	EG: 31 ± 3.5 CG: 31 ± 3.5	EG: 12.3 ± 9.8 CG: 12.3 ± 9.8	EG: 3×/week, 6 upper-body exercises and 4 lower-body exercises (maximum 12 reps), 3–7 sets, total 40–50 sets, rest interval: 30 s–2 min; 60 min, supervised heavy resistance training concentrating on the strengthening of major muscle groups CG: rest	10 weeks	HbA1c ↓ significantly in EG Total cholesterol ↓ significantly in EG Blood glucose ↓ significantly in EG Triglyceride, LDL-C ↓ in EG HDL-C  in EG Weight  in both Strength, endurance ↑ significantly in EG	2
Fuchsjager-Maryle et al. 2002 [32]	EG: 18 (11 F; 7 M) CG: 8 (3 F; 5 M)	EG: 42 ± 10; CG: 33 ± 11	EG: 20 ± 10; CG: 20 ± 10	EG: 2×/week at weeks 1–2; 3×/week at weeks 3–16; stationary cycling with increasing resistance till 60–70% HR; 3–5 min WU + 40 min + 5 min CD, supervised exercise CG: no training intervention	16 weeks	Weight  in EG BMI  in EG VO_2max_ ↑ significantly in EG HbA1c, LDL-C, HDL-C, triglycerides  in EG Total cholesterol tended ↓ in EG Daily insulin dosage ↓ significantly in EG Isometric muscle strength of both legs ↑ significantly in EG	3
Gusso et al. 2017 [25]	EG: 38 CG: 15	EG: 15.6 ± 1.3; CG: 15.5 ± 0.9	EG: 5.4 ± 3.4; CG: 7.5 ± 4.0	EG: 4×/week; (3×aerobic sessions + 1×resistance)/week at weeks 1–12, 4× circuit training/week at weeks 12–20; aerobic exercise: progressively to 85% HRmax at weeks 1–4; 85% HRmax at weeks 5–20; 40 min/session; resistance training: weight training and core exercise; 60 min exercise sessions per week (including WU and CD), supervised exercise CG: no training intervention	20 weeks	Body weight ↑ significantly in both groups Body fat percentage ↓ in EG, ↑ in CG Fat-free mass ↑ in EG HbA1c  in both groups VO_2peak_ ↑ in EG DBP (resting) ↓ significantly in EG Daily insulin dosage ↓ in EG	7
Heyman et al. 2007 [33]	EG: 9 (F) CG: 7 (F)	EG: 15.9 ± 1.5; CG: 16.3 ± 1.2	EG: 6.3 ± 4.4; CG: 8.4 ± 4.5	EG: 2×/week; 80–90% of HRR (measured by monitors); one 2-h supervised session + one 1-h unsupervised session, combined aerobic and strength sessions in ratio of 2:1 CG: spent equal amount of time on activities that did not require physical effort	24 weeks	Body weight ↑ significantly in both groups Fat mass  in EG, tended ↑ in CG Fat-free mass ↑ significantly in EG,  in CG Daily insulin dosage  in both groups PWC_170_ (aerobic fitness) in watts ↑ significantly in EG Total cholesterol, LDL-C, HDL-C, triglycerides  in both groups Quality of life (DQOL) ↑ in EG	4
Huttunen et al. 1989 [34]	EG: 16 CG: 16	EG: 11.9 (8.2–16.9) CG: 11.9 (8.2–16.9)	EG: 4.7 (0.6–12.0); CG: 5.6 (2.0–13.1)	EG: 1×/week, HR >150 bpm, 60 min, aerobic exercise (jogging, running, gymnastics and various kinds of active games) CG: continued with normal PA behavior	13 weeks	VO_2peak_ ↑ significantly in EG,  in CG HbA1c ↑ significantly in EG	5
Laaksonen et al. 2000 [35]	EG: 20 CG: 22	EG: 32.5 ± 5.7; CG: 29.5 ± 6.3	EG: 13.8 ± 9.2; CG: 10.8 ± 5.8	EG: 3×/week, 50–60% VO_2peak_, 20–30 min, at week 1, gradually increased to 4–5×/week, 60–80% VO_2peak_, 30–60 min at weeks 2–16, aerobic training CG: continued with normal PA behavior	12–16 weeks	VO_2peak_ ↑ significantly in EG; HbA1c, daily insulin dosage, BMI, body fat percentage  in both groups Total cholesterol, LDL-C ↓ in EG; HDL-C ↑ in CG; Triglycerides changes significantly greater in EG than CG	4
Landt et al. 1985 [36]	EG: 9 (6 F; 3 M) CG: 6 (2 F; 4 M)	EG: 16.1 ± 0.8; CG: 15.9 ± 0.3	EG: 6.7 ± 1.1; CG: 7.7 ± 1.5	EG: 3×/week, HR ≥160 bpm, 10 min WU + 25 min aerobic movement + 10 min CD, supervised exercise CG: continued with normal PA behavior	12 weeks	Daily insulin dosage  in both VO_2max_ ↑ in EG Lean body mass ↑ in EG Insulin sensitivity ↑ in EG HbA1c  in both groups	4
Maggio et al. 2011 [37]	EG: 15 (7 F; 8 M) CG: 12 (7 F: 5M)	EG: 10.5 ± 2.0 CG: 10.5 ± 2.9	EG: 3.1 ± 2.7 CG: 3.4 ± 1.7	EG: 2×/week; HR ≥140 bpm, 10 min WU + 10 min drop jump (height of platform from 20 cm for the first 3 months to 40 cm in last 6 months) + 60 min weight-bearing activities + 10 min CD; weight-bearing: ball games, jumping, rope skipping, and gymnastics supervised exercise CG: relatively inactive	36 weeks	BMI  in both groups Boy weight  in both groups	8
Newton et al. 2009 [38]	EG: 38 (16 F; 22 M) CG: 40 (20 F; 20 M)	EG: 11–18 years CG: 11–18 years		EG: wore open pedometer daily with a goal of 10,000 steps/day and received weekly text messages (reminding them to wear pedometer and be active) CG: received usual care for 12 weeks	12 weeks	HbA1c  in both groups SBP  in both groups DBP  in both groups BMI Z score  in both groups Quality of life (SQOL)  in both groups	7
Perry et al. 1997 ^a^ 666crossover [39]	EG: 31 CG: 31	EG: 41.5 ± 11; CG: 41.5 ± 11	EG: 14.1 ± 11.9; CG: 14.1 ± 11.9	EG: intensive lifestyle education with ≥3×/week, intensity and duration were based on individual fitness level and goals (walking, cycling, running, weight training); non-supervised and individualized aerobic PA CG: received standard care	24 weeks	Weight ↓ significantly in EG HbA1 ↓ not significantly in EG Triglycerides, total cholesterol, LDL-C  in EG HDL-C ↑ in EG VO2max ↑ in EG BP  in EG	5
Perry et al. 1997 ^b^ [39] crossover	EG: 30 CG: 30	EG: 42.8 ± 12.6; CG: 42.8 ± 12.6	EG: 16.8 ± 13; CG: 16.8 ± 13	EG: intensive lifestyle education with ≥3×/week, intensity and duration were based on individual fitness level and goals (walking, cycling, running, weight training); non-supervised and individualized aerobic PA CG: received standard care	24 weeks	HbA1 ↓ not significantly in EG BP  in EG Total, LDL-C ↓ significantly in EG Total, HDL-C  in EG	5
Roberts et al. 2002 [40]	EG: 12 CG: 12	14 ± 1.2	5.0 ± 3.1	EG: 3×/week, 45 min/session, HR ≥ 160 bpm, aerobic and anaerobic component in a ratio of 7:3; activities: running, light training circuits, games and aerobics; supervised exercise CG: unsupervised training	12 weeks	Aerobic power score ↑ significantly in EG,  in CG HbA1  in both groups BMI  in both groups Body mass  in both groups	3
Rowland et al. 1985 crossover [41]	EG: 14 CG: 14	(9–14)	4.2 (0.5–9.5)	EG: 3×/week, 10 min stretching + 20 min alternating 5-min walking/running increased to 30 min running 60% of HRR (160 bpm) + 5 min CD; recreational swim for 15 min twice weekly CG: rest	12 weeks	VO_2max_ ↑ in EG HbA1c  in both groups	6
Salem et al. 2010 ^a^ [2]	EG: 75 CG: 48	EG: 14.7 ± 2.2; CG: 15 ± 2.35	EG: 3.6 ± 1.8; CG: 4.9 ± 1.9	EG: 1×/week, 65 min; supervised exercise (1) Aerobic exercise (cycling/treadmill) at 65–85% HR_max_ (2) Anaerobic exercise (treadmill interval running at 85–95% HR_max_ for 1–2 min) (3) Leg extension and leg curl exercises (progressive resistive exercises, 10RM) (4) Different free strength and endurance exercises (10 min, 10 reps/set, number of sets increasing gradually) (5) Flexibility exercises (5 min stretching) (6) Neuromuscular exercises (5 min coordination exercises, 10 reps balance exercise regime on firm surface for 10 min, 10 reps which increased from 1 set to 3 sets after 6 sessions. CG: continued with normal PA behavior	24 weeks	HbA1c ↓ significantly in EG BMI (SDS) ↓ significantly in EG,  in CG Daily insulin dosage ↓ significantly in EG HDL-C ↑ significantly in EG,  in CG Triglycerides, total cholesterol, LDL-C ↓ in EG	4
Salem et al. 2010 ^b^ [2]	EG: 73 CG: 48	EG: 14.5 ± 2.4; CG: 15 ± 2.35	EG: 5.5 ± 2; CG: 4.9 ± 1.9	EG: 3×/week, 65 min; supervised exercise (1) Aerobic exercise (cycling/treadmill) at 65–85% HR_max_ (2) Anaerobic exercise (treadmill interval running at 85–95% HR_max_ for 1–2 min) (3) Leg extension and leg curl exercises (progressive resistive exercises, 10RM) (4) Different free strength and endurance exercises (10 min, 10 reps/set, number of sets increasing gradually) (5) Flexibility exercises (5 min stretching) (6) Neuromuscular exercises (5 min coordination exercises, 10 reps balance exercise regime on firm surface for 10 min, 10 reps which increased from 1 set to 3 sets after 6 sessions. CG: continued with normal PA behavior	24 weeks	HbA1c ↓ significantly in EG Daily insulin dosage ↓ in EG BMI (SDS) ↓ significantly in EG,  in CG DBP percentile ↓ in EG Triglycerides, total cholesterol, LDL-C HDL-C ↑ significantly in EG,  in CG	4
Stratton et al. 1987 [42]	EG: 8 (4 F; 4 M) CG: 8	EG: 15.1 ± 1.2 CG: 15.5 ± 0.9	EG: 3.7 ± 2.1; CG: 5.5 ± 3.3	EG: 3×/week, 30–45 min of supervised highly aerobic activities (treadmill jogging, cycle ergometer) on 2 of the 3 days; on 1 of the 3 days, activities: basketball, swimming, or resistance exercise machines (mostly aerobic); diet advice once a week, supervised exercise CG: were encouraged to exercise and given an outline exercise program	8 weeks	Daily insulin dosage ↓ in EG Bruce treadmill time, submaximal exercise heart rates ↑ in EG HbA1c, total cholesterol, triglycerides, HDL-C  in both groups	6
Tunar et al. 2012 [43]	EG: 17 (11 F; 6 M) CG: 14 (5 F; 9 M)	EG: 14.2 ± 2.2; CG: 14.3 ± 1.8	EG: 5.3 ± 4.1; CG: 6 ± 4.2	EG: 3×/week, 45 min, mat-based Pilates CG: continued with normal PA behavior	12 weeks	BMI SDS  in both groups HbA1c, daily insulin dosage  in both groups HDL-C ↑ in CG LDL-C, total cholesterol, triglycerides  in both groups Peak power, mean power, flexibility, vertical jump ↑ significantly in EG	5
Wallberg-Henriksson et al. 1986 [44]	EG: 6 (F) CG: 7 (F)	EG: 36 ± 2 (SEM); CG: 35 ± 2 (SEM);	EG: 14 ± 4 (SEM); CG: 13 ± 2 (SEM)	EG: 7×/week, 5 min low intensity WU + 15 min high intensity cycling at 60–70% VO_2max_ for first month, 70–80% VO_2max_ for the second and third month, 75–90% VO_2max_ for last 2 months) CG: continued with normal PA behavior	20 weeks	VO_2max_ ↑ significantly in EG HbA1c  in both groups Total cholesterol ↓ in both groups LDL-C, total triglycerides, blood glucose, HDL-C  in both groups	4
Wong et al. 2011 [45]	Home-based EG: 12 Self-directed EG: 5CG:11	Home-based EG: 11.62 ± 2.12; Self-directed EG: 13.44 ± 2.23 CG: 12.77 ± 1.79	Home-based EG: 4.42 ± 2.58; Self-directed EG: 3.42 ± 3.48; CG: 3.82 ± 2.87	Home-based EG: 3×/week, 10–30% HRR during WU and CD; 40–60% HRR during aerobic exercises; 10–20 min at week 1 to 20–30 min at weeks 3–12, delivered via VCR and/or phone; aid compliance and a handbook to provide guidance and exercise log, Self-directed EG: self-directed exercise CG: continued with normal PA behavior	12 weeks	HbA1c  in home-based EG, self-directed EG, CG at each study point; Self-directed EG had higher HbA1c than home-based EG; CG had higher HbA1c than self-directed EG at 9-month follow-up; VO_2peak_  in all groups	6
Yki-Jarvinen et al. 1984 [46]	EG: 7 (1 F; 6 M) CG: 6 (2 F; 4 M)	EG: 26 ± 1; CG: 24 ± 1	EG: 7 ± 1; CG: 9 ± 1	EG: 4×/week, 150–160 bpm, 60 min (4 × 15 min with 5-min rest intervals) CG: sedentary activity	6 weeks	VO_2max_ ↑ significantly in EG HbA1c  in EG Daily insulin dosage ↓ significantly in EG,  in CG Triglycerides, total cholesterol, HDL-C, LDL-C 	1

SD = standard deviation; SEM = standard error of the mean; EG = exercise group; CG = control group; F = female; M = male; PA = physical activity; min *=* minutes; s = seconds; WU = warm up; CD = cool down; HR_max_ = maximum heart rate; HR = heart rate; HRR = heart rate reserve; bpm = beats per minute; RM = repetition maximum; reps = repetitions; BMI = body mass index; SDS = standard deviation score; HbA1c = glycated hemoglobin; Fasting blood glucose levels = FBG; SBP = systolic blood pressure; DBP = diastolic blood pressure; VO_2max_ = maximal oxygen uptake; VO_2peak_ = peak oxygen uptake; PWC _170_ = peak oxygen uptake and physical work capacity at a heart rate of 170 beats/min; HDL-C = high-density lipoprotein cholesterol; LDL-C = low density lipoprotein cholesterol; DQOL = diabetes quality-of-life; SF-36 = Short Form (36) Health Survey; DQOL = Diabetes Quality-of-life; SQOL= Subjective Quality of Life; PEDro = Physiotherapy Evidence Database; ↑ = increase; ↓ = decrease; 

 = no changes; ^a^, ^b^ data in the study included more than one condition.

**Table 2 jcm-08-00253-t002:** Subgroup analyses of moderator variables of maximal aerobic power, HbA1c, daily insulin dosage, and total cholesterol.

Outcome	Moderator Variable	Subgroups	No. of Trials	No. of Participants	Pooled Meta-Analysis			Heterogeneity			Subgroup Differences		
					Mean Difference	95% (Confidence Interval)	*p* (Overall Effect)	*I*^2^ (%)	Chi^2^	*p*	*I* ^2^	Chi^2^	*p*
Maximal Aerobic Power (mL·kg^−1^·min^−1^)		Total	11	303	3.01	(0.94 to 5.07)	<0.01	80	50.68	<0.01			
	Age groups	Children and adolescents	6	161	2.98	(0.96 to 5.00)	<0.01	0	2.54	0.77	0.0	0.02	0.89
		Adults	5	142	3.24	(0.10 to 6.38)	0.04	91	46.08	<0.01			
	Exercise frequency	≥3 times/week	9	223	4.45	(3.37 to 5.52)	<0.01	0	6.48	0.59	97.7	44.01	<0.01
		<3 times/week	2	80	−0.17	(−1.01 to 0.67)	0.69	0	0.19	0.69			
	Type of exercise	Aerobic exercise	9	202	4.25	(3.16 to 5.34)	<0.01	0	7.36	0.50	0.0	0.43	0.51
		Combined aerobic and resistance training	2	101	2.32	(−3.36 to 8.00)	0.42	85	6.68	0.01			
	Program duration	>12 weeks	4	134	5.05	(3.81 to 6.29)	<0.01	0	2.22	0.53	96.7	30.44	<0.01
		≤12 weeks	7	169	0.48	(−0.37 to 1.53)	0.37	5	6.34	0.39			
HbA1c (%)		Total	24	862	−0.45	(−0.73 to −0.17)	<0.01	76	97.69	<0.01			
	Age groups	Children and adolescents	17	595	−0.60	(−1.07 to −0.14)	0.01	74	61.03	<0.01	72.4	3.62	0.06
		Adults	7	267	−0.10	(−0.33 to 0.13)	0.40	54	13.06	0.04			
	Exercise frequency	≥3 times/week	16	561	−0.53	(−0.88 to −0.17)	<0.01	69	47.64	<0.01	0	0.27	0.60
		<3 times/week	8	301	−0.34	(−0.93 to 0.24)	0.25	85	46.91	<0.01			
	Type of exercise	Aerobic exercise	14	316	−0.39	(−0.84 to 0.06)	0.09	74	50.36	<0.01	0	2.29	0.51
		Combined aerobic and resistance training	8	499	−0.56	(−1.05 to −0.08)	0.02	84	43.66	<0.01			
		Resistance training	1	16	−1.10	(−2.25 to 0.05)	0.06	-	-	-			
		Pilates	1	31	−0.10	(−1.08 to 1.28)	0.87	-	-	-			
	Program duration	>12 weeks	14	620	−0.56	(−0.95 to −0.17)	<0.01	83	76.76	<0.01	40	1.67	0.20
		≤12 weeks	10	242	−0.19	(−0.59 to 0.22)	0.38	27	12.35	0.19			
Daily Insulin Dosage (U·kg^−1^·day^−1^)		Total	6	355	−0.88	(−1.27 to −0.48)	<0.01	98	276.52	<0.01			
	Age groups	Children and adolescents	4	297	−1.69	(−2.43 to −0.95)	<0.01	99	256.38	<0.01	94.4	17.81	<0.01
		Adults	2	58	−0.09	(−0.19 to 0.02)	0.11	56	2.25	0.13			
	Exercise frequency	≥3 times/week	4	210	−1.54	(−2.21 to −0.88)	<0.01	99	270.57	<0.01	93.1	14.44	<0.01
		<3 times/weeks	2	145	−0.19	(−0.40 to 0.01)	0.07	83	5.71	0.02			
	Type of exercise	Aerobic exercise	3	80	−0.09	(−0.15 to −0.03)	<0.01	12	2.26	0.32	99.2	253.33	<0.01
		Combined aerobic and resistance training	2	244	−0.40	(−0.60 to −0.20)	<0.01	76	4.22	0.04			
		Pilates	1	31	−7.7	(−8.65 to −6.75)	<0.01	-	-	-			
	Program duration	>12 weeks	5	324	−0.20	(−0.35 to −0.06)	<0.01	89	34.82	<0.01	99.6	234.66	<0.01
		≤12 weeks	1	31	−7.70	(−8.65 to −6.75)	<0.01	-	-	-			
Total Cholesterol (mmol L^−1^)		Total	15	588	−0.38	(−0.71 to −0.04)	0.03	89	125.08	<0.01			
	Age groups	Children and adolescents	7	343	−0.84	(−1.22 to −0.46)	<0.01	75	24.32	<0.01	92.4	13.08	<0.01
		Adults	8	245	−0. 02	(−0.25 to 0.21)	0.86	57	16.31	0.02			
	Exercise frequency	≥3 times/week	13	451	−0.25	(−0.54 to 0.03)	0.08	80	59.51	<0.01	71.1	3.46	0.06
		<3 times/week	2	137	−0. 96	(−1.65 to −0.27)	<0.01	81	5.21	0.02			
	Type of exercise	Aerobic exercise	9	190	−0.10	(−0.37 to 0.16)	0.43	66	23.75	<0.01	62.5	5.34	0.07
		Combined aerobic and resistance training	5	382	−0.71	(−1.15 to −0.27)	<0.01	84	24.67	<0.01			
		Resistance training	1	16	−0.30	(−1.28 to 0.68)	0.55	-	-	-			
	Program duration	>12 weeks	10	498	−0.40	(−0.76 to −0.04)	0.03	90	89.96	<0.01	0.0	0.07	0.78
		≤12 weeks	5	90	−0.29	(−0.95 to 0.37)	0.38	58	9.49	0.05			

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
