# Peer review of "Cardiovascular Health Benefits of Exercise Training in Persons Living with Type 1 Diabetes: A Systematic Review and Meta-Analysis"

_jcm, 2019, doi:10.3390/jcm8020253_

Reviewer 1 Report

Abstract

Statement

Exercise is one of the cornerstones. Over the top. State exercise is advocated.

Glucose homeostasis: this is a term you shouldn’t use with T1d it can confuse. Suggested ‘glycaemic control (lower insulin dose hba1c)’ – there is debate around hba1c and daily glucose excursions – which are independent from each other, so state Hba1c is what you refer.

 Intro

Line 29 Clarify that the beta-cells are indeed lost, not just dysfunctional

Line: mention associative studies of PA and mortality, and there is limited RCT data

Moy CS, Songer TJ, LaPorte RE, Dorman JS, Kriska AM, Orchard TJ, Becker DJ, Drash AL: Insulin-dependent diabetes mellitus, physical activity, and death. Am J Epidemiol 1993;137:74-81

 Discussion:

Within each marker assessed you need to provide some inference or discussion point on statistical significance versus clinically meaningful.

With regards VO2, you mention that higher BMI patients at entry may experience a fall. Please how have you adjusted the fact if people lose weight the relative VO2 increases. This needs discussion, again, and points to clinically meaningless changes and again, over selling exercise. This needs to be clarified.

 Discussion point:

You need to provide a section detailing when the studies were completed. E.g. those before 2000, chances of the patients being on modern insulins is less, after 2000 to present, we have pumps, real time CGM. Thus it is unclear of the role of exercise when they highest standards of diabetes care are present, and no inference can be made.

You need to mention about hypoglycaemia risk, lack of data on incidence, and how this may impact future CVD – please explore recent papers on hypoglycaemia and cardiovascular events.

You should have a statement point on the number of trials that accurately demonstrated they had patients with T1D in the trial, providing C-peptide, autoantibody, or diagnosis history of the patients. There is likely a significant number of studies where not all of the patients are truly T1D.

Author Response

We appreciate greatly the considerate and insightful comments of each reviewer. We have addressed each of these comments in our revised manuscript and feel that this has strengthened the work. Our detailed responses to each reviewer are provided in following document. We have used tracked changes in our document to outline the changes and have also highlighted the changes in this response letter (as outlined in “red”).

Reviewer 2 Report

Authors performed a metanalysis of the cardiovascular improvements of T1DM patient after exercise training. The data have been properly analyzed, compared and summarized. However, some issues can improve the quality of the work.

-       Exercise training must be detailed in the manuscript. Different approaches have got different results. The intervals, oxygen consume, strength and resistance influence in the outcomes.

-       A briefly description of the cardiovascular alterations in T1DM is advised. In particular, the cardiac abnormalities, which may be related with a metabolic enhancement associated to exercise practice. Suggested reference: Fuentes-Antrás J et al 2015

-       A subsection in discussion describing the relationship between the parameters and the cardiovascular evolution, could be interesting. Thus, the intensive control of hyperglycemia and dyslipemia has not always given a positive result in patients. Also, the obesity paradox point out to different management of this pathology for cardiovascular prognosis in patients. A summary of what are the recommendations for those parameters, including benefits and drawbacks of their regulation, is required

-       Introduction: Young people may not be ranged between 8 and 43 y-o

Author Response

(The authors gave the same response as above.)
